# The Non-JAZ TIFY Protein TIFY8 of *Arabidopsis thaliana* Interacts with the HD-ZIP III Transcription Factor REVOLUTA and Regulates Leaf Senescence

**DOI:** 10.3390/ijms24043079

**Published:** 2023-02-04

**Authors:** Ana Gabriela Andrade Galan, Jasmin Doll, Svenja Corina Saile, Marieluise Wünsch, Edda von Roepenack-Lahaye, Laurens Pauwels, Alain Goossens, Justine Bresson, Ulrike Zentgraf

**Affiliations:** 1Center for Plant Molecular Biology (ZMBP), University of Tübingen, Auf der Morgenstelle 32, 72076 Tübingen, Germany; 2Department of Plant Biotechnology and Bioinformatics, Ghent University, 9052 Ghent, Belgium; 3Center for Plant Systems Biology, VIB, 9052 Ghent, Belgium

**Keywords:** TIFY8, REVOLUTA, transcription factor regulation, leaf senescence, *Arabidopsis thaliana*, jasmonic acid signaling, PEAPOD, JAZ proteins

## Abstract

The HD-ZIP III transcription factor REVOLUTA (REV) is involved in early leaf development, as well as in leaf senescence. REV directly binds to the promoters of senescence-associated genes, including the central regulator *WRKY53*. As this direct regulation appears to be restricted to senescence, we aimed to characterize protein-interaction partners of REV which could mediate this senescence-specificity. The interaction between REV and the TIFY family member TIFY8 was confirmed by yeast two-hybrid assays, as well as by bimolecular fluorescence complementation in planta. This interaction inhibited REV’s function as an activator of *WRKY53* expression. Mutation or overexpression of *TIFY8* accelerated or delayed senescence, respectively, but did not significantly alter early leaf development. Jasmonic acid (JA) had only a limited effect on *TIFY8* expression or function; however, REV appears to be under the control of JA signaling. Accordingly, REV also interacted with many other members of the TIFY family, namely the PEAPODs and several JAZ proteins in the yeast system, which could potentially mediate the JA-response. Therefore, REV appears to be under the control of the TIFY family in two different ways: a JA-independent way through TIFY8, which controls REV function in senescence, and a JA-dependent way through PEAPODs and JAZ proteins.

## 1. Introduction

Leaf polarity, polarity along the shoot–root axis, and stem cell specification and proliferation are regulated by class III homeodomain leucine zipper (HD-ZIP III) transcription factors [1,2,3]. However, these factors, namely REVOLUTA (REV), not only control these critical steps in early plant development, but are also involved in later steps of development such as leaf senescence and reproduction [4,5,6]. Chromatin-IP revealed that REV binds directly to the promoter of the senescence-associated transcription factor *WRKY53*, which is one of the hubs of senescence regulation and is tightly controlled on multiple levels [4,7]. REV appears to be an important driver for senescence, as the delayed-senescence phenotype of the *rev5* mutant is stronger than that of the *wrky53* mutant. This can be easily explained by the fact that several other senescence-associated genes (SAGs) are direct targets of REV, as shown by a ChIP-Seq experiment [4,5,8]. Vice versa, there is little indication that WRKY53 plays a role in early leaf development and leaf polarity. Accordingly, the interaction of REV and *WRKY53* appears to be dependent on the developmental stage, and the influence of REV on *WRKY53* expression seems to be most efficient at the onset of monocarpic senescence. One possible explanation could be that REV needs specific modifications or specific interaction partners to act as an activator of *WRKY53* expression and senescence. We can already show that the redox state of the REV protein can alter its binding to the *WRKY53* promoter, in which the reduced form binds more efficiently to the DNA [4]. However, on the other hand, *WRKY53* expression is induced by increasing levels of hydrogen peroxide, and the REV protein is somehow involved in this response, as the amplitude of this response is dampened in the *rev5* mutants. This is a contradiction that has not yet been solved and points to a more complex scenario. Moreover, the binding preference of REV to the different binding-sites in the *WRKY53* promoter changes during the progression of development and senescence, also indicating an adaptation of the REV properties to certain developmental stages [4]. 

On the molecular level, the expression of HD-ZIP III transcription factors is controlled by miRNA165/166, and the functionality of the proteins is controlled by the interaction with mircoProteins called LITTLE ZIPPER [9,10]. A leucin zipper domain mediates homo- and heterodimerization between HD-ZIP III factors as well as interaction with the LITTLE ZIPPER proteins, which then block the dimerization and the function [10]. In addition, the C-terminal MEKHLA domain of the HD-ZIP III factor REV was suggested to inhibit dimerization through a steric masking mechanism [11]. In general, PAS domains are sensor domains that respond to a variety of chemical and physical stimuli that regulate a wide range of signal transduction pathways [12]; however, the redox sensitivity of REV is not mediated by the PAS domain [4]. Already in 2013, Reinhart and colleagues characterized proteins that could interact with the full-length REV protein and a truncated version that lacks the MEKLHA domain using the yeast two-hybrid system [13]. One of these proteins was TIFY8, a non-canonical member of the TIFY family.

TIFY proteins are defined by a highly conserved TIFY motif (TIF[F/Y]XG) that resides within the larger ZIM domain (Zinc-finger protein expressed in Inflorescence Meristem). In Arabidopsis, TIFY proteins are represented by 18 family members which are subdivided into two classes due to the presence or absence of a C2C2-GATA domain. Three members that have this C2C2-GATA domain belong to class I, whereas the fifteen other members lack this domain. Twelve of the class II members are so-called JAZ proteins. JAsmonate ZIM-domain (JAZ) proteins are repressors that prevent the action of multiple transcription factors which execute the response of the plant to the hormone jasmonic acid (JA). Most JAZ proteins use the TIFY/ZIM domain to interact with an adapter protein called NINJA (NOVEL INTERACTOR OF JAZ). The NINJA protein contains an EAR (ERF-associated Amphiphilic Repression) motif to recruit the repressor TOPLESS [14,15,16]. JAZ-bound transcription factors are inactive due to the presence of TOPLESS in the complex, but the transcription factors can be activated rapidly in response to jasmonoyl-L-isoleucine (JA-Ile), the bioactive form of JA. JA-Ile can bind to the Jas domain of the JAZ proteins, thereby acting as a “molecular glue” between the JAZ proteins and CORONATINE-INSENSITIVE1 (COI1) [17]. COI1 represents the F-Box protein of the E3-ubiquitin ligase complex SCF^COI1^ [18], which directs the JAZ proteins to 26S-mediated proteasomal degradation in the presence of JA-Ile.

In contrast to the JAZ protein, the TIFY proteins PEAPOD1 (PPD1) and PEAPOD2 (PPD2) contain a divergent C-terminal Jas domain and an additional N-terminal PPD-domain. PEAPOD proteins control lamina size and curvature in Arabidopsis [19] and are negative regulators of meristematic proliferation to control organ size [20]. It was recently described that PEAPOD repressors modulate and coordinate developmental responses to light intensity [21]. For PPD1 and PPD2, TOPLESS can be recruited by KIX8 (KINASE-INDUCIBLE DOMAIN INTERACTING8) or KIX9 instead of NINJA [22]. TIFY8, which is even more divergent, completely lacks a Jas domain for the response to JA-lIe but still contains a ZIM domain [23]. In accordance with the lack of the Jas domain, TIFY8´s stability is not affected by JA treatment. No other specific protein domains other than the ZIM domain have been described so far; moreover, no direct DNA-binding could be observed. Despite the functional ZIM domain, no interaction with the JAZ proteins was found; only the two PEAPOD proteins PPD1 and PPD2 as well as NINJA were able to interact via the ZIM domain of Arabidopsis TIFY8. KIX8 and KIX9 were also identified in Arabidopsis TIFY8 protein complexes [23]. Therefore, we hypothesized that TIFY8 could recruit the repressor TOPLESS via NINJA or KIX8/9 to the REV complex and counteracts REV function as an activator of transcription.

Here, we analyzed the interaction interface between REV and TIFY8 in more detail. Using the yeast-two-hybrid system, we could demonstrate that REV interacts with the ZIM domain of TIFY8. However, the ZIM domain is necessary but not sufficient for REV-binding; the surrounding regions as well as the C-terminal residues are also involved. The REV and TIFY8 interactions were confirmed in planta, using Arabidopsis protoplasts and *Nicotiana benthamiana* leaves. Moreover, we analyzed the impact of TIFY8 on senescence regulation in *tify8* mutants and *TIFY8* overexpression (*TIFY8*-OE) plants. Here, a clear repressing effect of TIFY8 on senescence was observed, indicating that the interaction between TIFY8 and REV might have a biological function. However, the effect on early leaf development was less pronounced, as leaf morphology was only slightly affected in the *tify8* mutants and *TIFY8*-OE plants.

Even though JA-Ile and the PPD proteins are known to be involved in the repression of cell proliferation and cell size during leaf growth [24], and although TIFY8 appears to have a function in the PPD/KIX repressor complex in Arabidopsis, the relationship between JA-Ile and PPD and TIFY8 function is still unclear. Therefore, we analyzed whether JA has a role in TIFY8-REV regulation.

## 2. Results

### 2.1. TIFY8 and REVOLUTA Proteins Physically Interact

In order to characterize the interaction of REV with TIFY8 in more detail, we analyzed the interaction of REV with the full length TIFY8, as well as with several truncated versions of TIFY8 in yeast cells using the Matchmaker^®^ Gold Yeast-Two-Hybrid (Y2H) System. As previously reported, TIFY8 has been identified as an interacting partner of REV in a screen using a yeast library prepared from mRNA of inflorescence meristems [13]. In this analysis, the presence of the MEKLHA domain of REV did not influence the interaction in the re-testing. Here, we tested whether REV interacts with the full length TIFY8 protein, with the C- and the N-terminal part of the protein, and with several truncated versions (Figure 1a). Mating was repeated 3–7 times, and yeast cells were grown on selective media with increasing stringency (Figure 1b). Expression of the fusion proteins in the yeast cells was confirmed by Western Blot and immune detection (Appendix A). One example of the yeast colonies´ growth on selection media is shown in Figure 1b, and a summary of the 3–7 mating approaches is displayed as a heat map in Appendix A. An interaction between REV and the full-length TIFY8 was observed in all matings, as well as an interaction with the C-terminal half of the protein (TIFY8-C) that included the TIFY/ZIM domain (Figure 1b and Appendix A). The N-terminal protein parts without the TIFY/ZIM domain (TIFY8-N, TIFY8-N1) were not able to bind to the REV protein even though we could clearly show that the truncated proteins were expressed in the yeast cells (Appendix A). To narrow down the interaction interface, we divided the C-terminal half into several subdomains C1 to C4. Here, only the constructs containing the TIFY/ZIM domain were able to interact with REV. However, the interaction between REV and the TIFY/ZIM domain alone (TIFY8-C4) appears to be only weak (Figure 1b and Appendix A). This clearly indicates that the TIFY/ZIM domain together with the flanking regions is the interaction interface between REV and TIFY8.

In order to confirm the interaction between REV and TIFY8 in planta, we used bimolecular fluorescence complementation assays (BiFC). Therefore, Arabidopsis protoplasts that were prepared from cell culture cells, as well as leaves of *Nicotiana bentamiana,* were transiently transformed with BiFC constructs bearing REV fused to one half of the yellow fluorescent protein (YFP) and TIFY8 or the TIFY8-C terminus and its truncations fused to the other half of the YFP. In addition, a red fluorescence protein (RFP) was encoded by the same construct to control for transformation efficiency. The fluorescent protoplasts were analyzed with the CYTOFLEX cell sorter, indicating the portion of protoplasts with BiFC of YFP compared to RFP-transformed cells. The infiltrated tobacco leaves were analyzed by confocal laser scanning microscopy.

In both systems, we could clearly confirm that the full length TIFY8 protein could interact with REV also in planta. The interaction takes place predominantly in the nucleus, as expected for a transcription factor (Figure 2). As in the yeast cells, two TIFY8-truncated versions containing the TIFY/ZIM domain (TIFY8-C, TIFY8-C1) were able to interact with REV, even though it appears that the interactions were not as efficient as with the full-length TIFY8. However, the TIFY8-C4 construct containing only the TIFY/ZIM domain, which was sufficient to interact with REV in yeast cells, appears not to be sufficient to produce a fluorescence signal in the BiFC in planta. Moreover, although TIFY8-C2 and TIFY8-C3 were not able to interact with REV in yeast cells, we observed that the C-terminal part that is downstream of the TIFY/ZIM domain is also able to contribute to the interaction with REV in plant cells. This could possibly indicate that in plant cells, an additional plant protein or a plant-specific modification is involved in the interaction. In conclusion, the TIFY/ZIM domain, as well as the C-terminal regions, is involved in the interaction with REV.

### 2.2. TIFY8 Inhibits the Transactivation Capacity of REV

To ascertain the consequences of REV/TIFY8 interaction on the regulatory function of REV, we tested the induction of the direct target gene *WRKY53* by REV in the absence or presence of TIFY8. Col-0 Arabidopsis protoplasts were transiently transformed with the reporter construct P*_WRKY53_*:*GUS* together with the effector constructs 35S:*REV*, 35S:*TIFY8,* or both in combination. Moreover, we aimed to investigate whether JA influences TIFY8/REV interaction. As expected, REV induced the expression of the promoter-*WRKY53*-driven reporter gene. In contrast, TIFY8 had a clear repressive effect on the *WRKY53* promoter. If both effector proteins were co-expressed in the protoplasts, an even stronger repressing effect became obvious. This non-additive effect and the fact that the repressing effect of TIFY8 is even enhanced when both proteins are highly expressed can only be interpreted that with increased amounts of REV, the repressor function of TIFY8 is more pronounced due to the direct interaction between them and the dominant repressing effect of TIFY8. To our surprise, JA had an enhancing effect on the induction of *WRKY53* expression by REV, whereas no influence on the repressor function of TIFY8 could be observed (Figure 3).

It has been previously described that in contrast to the other class II TIFY proteins, TIFY8 stability is not regulated by JA, as it also lacks a Jas domain for JA–Ile binding [23]. However, it has not been observed before that REV function can be enhanced by JA. The JA effect is most likely mediated on the protein level, as in this case, *REV* expression is driven by the 35S cauliflower mosaic virus promoter, which is not sensitive to JA. In addition, we used a REV construct which is no longer responsive to miRNA165/166 (REVd) [10] so that the JA effect is not related to miRNA expression.

### 2.3. Expression Pattern of REV and TIFY8

If the interaction between TIFY8 and REV is of biological relevance, a temporal and spatial overlap in their expression patterns should be observed. As it was already shown that REV has a positive regulatory effect on leaf senescence [4,5,6], we tested leaf tissue of different developmental stages for the expression of *TIFY8* and *REV*—as well as the direct target gene of REV, *WRKY53*—using qRT-PCR. *ACTIN2* was used as a reference gene. As expected for a regulator and its direct target gene, the *REV* and *WRKY53* expression patterns were very similar. In the youngest plants analyzed here (4-week-old plants), *REV* expression and *WRKY53* expression were low, but both genes were upregulated at the onset of monocarpic senescence in leaves of 6- and 7-week-old plants. In contrast, *TIFY8* had its lowest expression in 6- and 7-week-old plants and its highest in 4-week-old plants when *REV* and *WRKY53* expression was low. This is consistent with the function of TIFY8 as a repressor of REV function. The expression pattern of *TIFY8* appears to ensure that in younger plants, senescence is not induced, and *WRKY53* and other direct senescence-associated target genes of REV are not activated. In contrast, in older plants, when *REV* expression increased and SAGs such as *WRKY53* were activated, *TIFY8* expression and its repressing effect on REV should be low. Interestingly, not only the amount of the *TIFY8* transcripts was reduced in older plants, but also the ratio between the two described splicing variants (Appendix A) was altered, and the portion of splicing variant 2 was decreased (Figure 4). As was the case for the JAZ proteins, the C-terminus is truncated in splicing variant 2; however, in the JAZ proteins, retention of the so-called Jas intron generates truncated proteins that lack C-terminal amino acids. These truncated JAZs retain the ability to interact with transcription factors, such as MYC2, but have a reduced capacity to form complexes with COI1 in comparison to their respective full-length isoforms. Therefore, these truncated splice variants of the JAZ proteins are dominant repressors of JA signaling and provide a general mechanism to reduce the fitness costs associated with over-stimulation of the JA signaling pathway [25]. Whether the TIFY8 splice variants also expand the functional repertoire still has to be elucidated.

### 2.4. Involvement of TIFY8 in Early Leaf Development and Senescence

To unravel the biological function of the REV/TIFY8 interaction, we analyzed early leaf development as well as monocarpic senescence in *tify8* mutants and a *TIFY8*-OE line. We used two previously characterized T-DNA insertion lines, *tify8-1T* and *tify8-2T,* and the highly overexpressing *TIFY8*-OE line 1 [23]. Moreover, we describe here two additional *TIFY8* alleles that we constructed using CRISPR/Cas9. As the T-DNA insertions were localized in the first exon and first intron, we hypothesized that these insertion lines might not be full KO [23]. Hence, we targeted *TIFY8* with two constructs, each with two sgRNAs [26], targeting exon 5 and either the sequence encoding the TIFY domain (exon 4) or exon 3 (Appendix A). Both lines are homozygous for indels at the targeted sites. All lines were grown side by side with the Col-0 wildtype plants and the *rev5* mutant line for comparison. The mutant lines and the overexpressor line have been analyzed in two separate experiments due to space limitations in the climate chambers and are, therefore, presented separately with their corresponding controls. The phenotyping was repeated several times with the same outcome, and one example is presented here.

First, the different colors of the leaves of one rosette were quantified using an automated colorimetric assay (ACA) tool (Figure 5a) which was developed in our lab [27]. In Figure 5, we show the results of only one of the mutant lines, *tify8-4*; the results of all four *tify8* mutant lines are presented in Appendix A. In addition, the phenotypical appearance of the leaves is shown in Appendix A; both pointing already to an acceleration of senescence in *TIFY8* loss-of-function lines, whereas the *TIFY8*-OE line showed delayed senescence more similar to the *rev5* mutant. (Figure 5a and Appendix A). However, the length of the main shoot as well as the number of side shoots was not significantly different in all lines except the *rev5* mutant, which had a shorter main shoot (Appendix A) but more side shoots (Appendix A), which was already described before [5]. This indicates that TIFY8 might alter predominantly the senescence effects of REV.

Moreover, additional parameters were analyzed to describe different changes in the complex process of senescence. Chlorophyll loss, functionality of photosystem II using pulse amplitude modulation (PAM) fluorometry, deterioration of the plasma membrane measured by ion leakage and lipid peroxidation (the latter also giving hints on oxidative stress), and senescence-associated gene (SAG) expression were monitored over time. For more reliable comparisons, leaves of defined positions within the rosette were used to analyze the different parameters in the same plant following the guidelines given in [27]. Leaf No. 5 and 10 were first used for PAM fluorometry, and subsequently, chlorophyll was extracted from Leaf No. 5. The loss of *TIFY8* function accelerated chlorophyll loss, as well as loss of the functionality of photosystem II (Figure 5b, Appendix A), which is consistent with the phenotypical appearance. Even though the differences were not significant except for *rev5*, all mutants had lower Fv/Fm values compared to Col-0 and *rev5* plants (Appendix A), whereas the *TIFY8*-OE displayed a delayed senescence phenotype similar to that of *rev5* (Figure 5 and Appendix A). Ion leakage was measured in leaf No. 4, and lipid peroxidation was analyzed in Leaf No. 9. Both ion leakage and lipid peroxidation were more pronounced in the *tify8* mutants and less pronounced in the *rev5* mutant and *TIFY8-OE* (Figure 5c and Appendix A).

In addition, expression of two senescence-associated marker genes, namely the cysteine protease *SAG12* and the short-chain alcohol dehydrogenase *SAG13*, was analyzed in Leaf No. 7 using qRT-PCR and again *ACTIN2* as a reference gene. *SAG12* and *SAG13* are expressed more highly in all *tify8* mutant lines in week 7 compared to Col-0 wildtype plants. In contrast, *TIFY8*-OE leaves had lower *SAG12* and *SAG13* expression than Col-0 leaves and behaved more similarly to the *rev5* mutants (Figure 5d, Appendix A).

In conclusion, TIFY8 acts as a repressor of leaf senescence, most likely through the interaction with REV. In contrast to the clear effects on senescence, the impact of TIFY8 appears to be less pronounced in the early stages of leaf development. The *rev5* mutant plants showed the clear downward curvature of the leaves as described before [8,10], whereas the Col-0 wildtype plants, as well as the *tify8* mutants, had flat leaves. The *TIFY8*-OE line also developed slightly downward-curled leaves, but this tendency was much less pronounced than in *rev5* mutants (Appendix A).

Taken together, TIFY8 has a regulatory impact on REV function but preferentially in late leaf developmental stages and less pronounced in early leaf development.

### 2.5. Impact of JA on the REV/TIFY8/WRKY53 Network

As the activation of the *WRKY53* promoter by REV was enhanced by JA in the transiently transformed protoplasts and JA was shown to be involved in senescence regulation [28,29], we aimed to investigate how JA influences the REV/TIF8/WRKY53 network. Therefore, we measured JA levels in the Col-0 wildtype plants, in two *tify8* mutants, and in the *TIFY8*-OE line, as well as in the *rev5* mutant. We could confirm that JA levels increased with the age of the plants, as already observed before [30]. However, there was no significant difference between the lines (Figure 6), indicating that there is no feedback regulation of REV or TIFY8 on JA biosynthesis.

Moreover, the expression of *REV*, *TIFY8*, and *WRKY53* was analyzed by qRT-PCR in the wildtype and the mutant lines after JA treatment. Short-term effects 6 h after treatment as well as long-term effects after 24 h and 96 h were determined. Two different developmental stages, namely 3- and 5-week-old plants, were analyzed to test whether there is an influence of the developmental stage on hormone response. The expression of the *REV*, *TIFY8*, and *WRKY53* genes are presented as heat maps relative to the expression levels of the respective genes after MOCK treatment (Figure 7).

In Col-0, there appears to be a difference between 3- and 5-week-old plants, as *REV* and *TIFY8* are more severely downregulated 6 h after treatment in 5-week-old plants. After 24 h in 5-week-old plants, there is a switch in the response; now *REV* and *TIFY8* are more strongly expressed after JA treatment, which is even intensified for *REV* after 96 h. This effect appears to be delayed in the 3-week-old plants. Here, the switch can be observed only after 96 h and is also less severe.

In contrast, *WRKY53* expression was slightly induced by JA treatment in 3-week-old plants after 6 h, then slightly reduced after 24 h, and increased again after 96 h. Again, this pattern changed in 5-week-old plants, in which the expression was slightly reduced 6 h after treatment and then increased 24 h after treatment, which was also intensified after 96 h. *REV* expression appears to be more responsive to JA than *TIFY8* and *WRKY53* expression, at least in 5-week-old plants, and the *WRKY53* pattern resembles the *REV* pattern more in 5-week-old plants.

This is consistent with the previously observed influence of REV on *WRKY53* expression during senescence, but a less clear connection between these partners during early leaf development appears to exist. From the expression pattern of all three genes in the *rev5* and *tify8* mutant lines, in which the leaf material of the two mutants *tify8-1T* and *tify8-4* has been combined, we can conclude that REV as well as TIFY8 are somehow involved in the regulation of each other, as the expression patterns change for *REV* in the *tify8* mutant and vice versa. Moreover, both genes appear to be part of a feedback regulation on their own expression. Again, this appears to be more pronounced in 5-week-old plants than in 3-week-old plants. Taken together, JA has short- and long-term effects on the expression of all three genes. The effects are different and more pronounced in 5-week-old plants, but REV and TIFY8 proteins appear to be involved in the response of all three genes to JA in a complex and developmentally dependent manner.

### 2.6. Is the JA Influence on REV Function Mediated by Interaction with Other Proteins of the TIFY Family?

It has become clear that TIFY8 interacts directly with REV and has a repressing function on REV. However, TIFY8 has no Jas domain and is not able to sense JA-Ile directly. Therefore, the question as to how the enhancement of the REV function by JA is mediated is still open. Thus, we evaluated whether REV can also interact with other members of the TIFY family. We first tested the two class II TIFY PEAPOD proteins, as TIFY8 can interact with these two proteins but not with all other JAZ proteins [23]. Moreover, the PEAPOD proteins are involved in the early stages of leaf development in the formation of flat leaves in Arabidopsis [31], which would fit very well with the REV function in early leaf development. Therefore, we analyzed the two PEAPOD proteins in the yeast two-hybrid system, and both were able to interact not only with TIFY8 but also with REV (Figure 8 and Appendix A).

For PPD2, we could also show with truncated versions that the TIFY/ZIM domain is necessary but not sufficient to mediate the association with REV. In this case, the N-terminal region with the PPD domain appears to be involved in addition to the TIFY/ZIM domain (Figure 8 and Appendix A).

In addition to the PEAPOD proteins, the JAZ proteins were tested for interaction with REV in the yeast two-hybrid system. REV was also able to interact with many other JAZ proteins, namely JAZ1, 2, 4, 5, 9, and 10, but not or almost not with JAZ3, 6, 7, 8, and 12 (Figure 9 and Appendix A), clearly indicating that REV appears to be controlled by JAZ proteins and JA signaling. However, how this selective interaction is mediated and whether this interaction is of biological relevance will be the subjects of further investigations.

## 3. Discussion

Under optimal and stress-free conditions, leaf senescence is governed by the age of the leaves and the whole plant [32]. It has developed to allow efficient usage of the resources of the plant for growth, as well as for storage for the sake of the next generation via a well-organized recycling program to remobilize carbon, nitrogen, and mineral resources out of the senescing tissue into the developing parts of the plant, such as new leaves or fruits and seeds. However, long-lasting unfavorable stress conditions, such as drought, salinity, and nutrient deficiency, lead to premature senescence as an exit strategy. This ensures the production of offspring even under such a barren environment. Premature senescence is often combined with a tradeoff in seed number and quality [33]. To integrate all kinds of stress responses into this developmental process, highly complex gene regulatory networks have to be in place [7]. Approximately one fourth of all Arabidopsis genes are differentially regulated during the onset and progression of senescence [30,32]. Detailed transcript profiling over 22 time points of a defined leaf of *Arabidopsis thaliana* during onset and progression of leaf senescence enabled researchers to build a distinct chronology of events [30]. Genes related to the regulation of intracellular ROS levels as well as genes involved in abscisic acid (ABA) and JA production and signaling are among the early induced transcripts, indicating that ROS, ABA, and JA are important early signals in leaf senescence. This fact is in agreement with a relatively early increase in JA levels in Leaf No. 7 in Arabidopsis rosettes shown by Breeze and colleagues [30], which was confirmed here in our studies (Figure 6). Likewise, an increase in intracellular hydrogen peroxide contents during the onset of monocarpic senescence has been described [34,35].

This massive reprogramming of the transcriptome implies a central function for transcription factors. Almost all transcription factor families in plants are involved in senescence regulation processes; however, the families of WRKY and NAC factors, which largely expanded in the plant kingdom, are overrepresented in the senescence transcriptome of Arabidopsis [36]. In contrast to systems biology approaches, we tried to understand these complex interactions starting from one of the regulatory hub proteins, namely WRKY53. Expression, activity, and degradation of the WRKY53 protein are tightly controlled, involving many feedback loops and double bottoms [7]. Moreover, a “leaf developmental memory” that links early developmental processes to leaf senescence appears to exist [4,7,37], and by this mechanism, if early development is somehow disturbed, senescence is delayed. The transcription factor REV appears to be part of this memory, as REV is involved in early developmental processes, such as the establishment of leaf polarity, lateral meristem initiation, or vascular development, but also directly regulates the expression of *WRKY53* during leaf senescence [4]. However, a function of WRKY53 in early development has not yet been described, indicating that the interaction of REV with the promoter of *WRKY53* appears to be dependent on the developmental stage. The preferential binding of REV to different *cis*-elements in the *WRKY53* promoter [4] points to the involvement of an additional factor driving this selectivity. Here, we could characterize the non-canonical TIFY protein TIFY8 as a possible regulator. TIFY8 can interact with REV in yeast cells and in planta (Figure 1 and Figure 2) and can block the inducing function of REV on the promoter of *WRKY53* (Figure 3). This is consistent with the function described for TIFY8 as a repressor of transcription via the interaction with NINJA and/or KIX8/9 function as adapters for TOPLESS, which mediates transcriptional repression [14,22]. Moreover, *TIFY8* is more highly expressed in leaves during early developmental stages before the onset of senescence, when REV induction of *WRKY53* expression should still be inhibited. In contrast, *REV* and *WRKY53* expression increase during the onset of senescence, whereas *TIFY8* expression is lowered again (Figure 4). If this is the case and TIFY8 is involved in REV repression as described, a loss of TIFY8 function would lead to early activation of REV function and, thereby, to accelerated senescence. This is exactly what could be observed in the T-DNA insertion lines, as well as in the CRISPR/Cas lines of *TIFY8*. Here, we clearly observed an accelerated loss of photosynthetic activity and chlorophyll content, an early expression of *SAG*s (Figure 5, Appendix A), and an earlier deterioration of the plasma membrane documented by higher ion leakage and higher lipid peroxidation rate (Figure 5 and Appendix A) in the loss-of-function mutant lines compared to wildtype plants or *rev5* mutants. In contrast, the *TIFY8-*OE line phenocopied the *rev5* mutant in all these aspects (Figure 5, Appendix A). Therefore, we concluded that TIFY8 has a role as a negative regulator of senescence, most likely through the inhibition of REV, which can activate direct senescence-associated target genes such as *WRKY53*.

We have identified the TIFY/ZIM domain of TIFY8 to be involved in the interaction with REV. However, in planta, the TIFY/ZIM domain is not sufficient for the contact, but additional regions in the C-terminal part of the protein are required (Figure 1 and Figure 2). Moreover, REV does not interact exclusively with TIFY8 of the TIFY family. The two PEAPOD proteins, PPD1 and PPD2, can also interact with REV, but in this case, the N-terminal PPD domain is required in addition to the TIFY/ZIM domain. Moreover, several but not all JAZ proteins are also able to interact with REV in yeast, indicating a certain selectivity that might be mediated by the additional regions, as the TIFY/ZIM domain is highly conserved between the TIFY proteins (Appendix A). The interactions with the JAZ proteins could explain why in the reporter assay with the P*_WRKY53_*:*GUS* construct, activation by REV was increased after JA treatment. JAZ proteins are degraded upon JA-Ile perception via SCF^COI1^ and the 26S proteasome [14]. TIFY8 and PEAPOD proteins are also able to interact with each other (Figure 8), whereas all other JAZ proteins were not able to interact with Arabidopsis TIFY8 [23], demonstrating clear differences between the PEAPOD and the JAZ proteins. JAZ proteins have been shown to recognize MYC transcription factors; this process occurs most likely via a conserved linear motif SL••FL•••R. However, PEAPOD proteins which lack this motif do not recognize MYC protein unless this motif is implemented into the proteins by mutagenesis [38]. MYC2-5 redundantly regulates JA-induced leaf senescence under the control of JAZ proteins [39]. So far, there are no indications that PEAPOD proteins regulate senescence, but they are involved in early leaf development in Arabidopsis to form a flat leaf [31]. Accordingly, PEAPODs could be involved in early developmental processes directed by REV, and TIFY8 could be involved in late developmental processes directed by REV. However, we still need to analyze the impact of PEAPOD proteins on REV function and senescence in more detail in the future.

The JA effect on the *REV*, *TIFY8,* and *WRKY53* expression appears to be complex and development-dependent. Short- and long-term effects can be different at different developmental stages (Figure 7). However, JA appears to have the highest effects on *REV* expression in 5-week-old plants. Here, JA could induce high *REV* expression after 96 h so that a long-term increase in JA, as it is observed during early senescence (Figure 6), could contribute to the increased expression of *REV,* as *REV* mRNA and JA levels increase in parallel. In the *tify8* mutants, as well as in the *rev5* mutant, differences in expression of *REV* and its direct target gene *TIFY8* could be observed, indicating that complex cross- and feedback regulation occur but with milder effects. However, JA levels appear to be not significantly different in all tested lines, indicating that there is no influence of TIFY8 or REV on the JA biosynthesis. Vice versa, an influence of JA on the REV activity was observed (Figure 3) which is most likely mediated by either the PEAPOD and/or the JAZ interactions with REV. In conclusion, a JA-independent regulation of REV via TIFY8 and a JA-dependent regulation of REV via PEAPOD and JAZ proteins appear to exist, and these regulation mechanisms might act in concert and/or in different developmental stages. A simplified model of this complex interplay is presented in Figure 10. Here, we speculated that the interaction between PPDs or JAZs with REV also inhibits the function of REV as an activator; however, the impacts of PPD and JAZ proteins on early and late developmental functions of REV will be the subject of future investigations. In addition, the genetic background of the JA-insensitive mutants *jar1* and *coi1* will be used to characterize the role of JA signaling in more detail.

## 4. Materials and Methods

### 4.1. Yeast Two-Hybrid Assays

The TIFY8, PEAPOD, and JAZ yeast constructs used were described before in [22,23]. The yeast strain Y2H Gold (mating type a; Clontech; Takara Bio Europe SAS, Saint-Germain-en-Laye, France) was transformed with the bait expressed from the pGBKT7 vector containing the GAL4 DNA-binding domain and the *TRP1* marker gene. The yeast strain Y187 (mating type α; Clontech) was transformed, with the preys expressed from the pGADT7 vector containing the GAL4 activation domain and the *LEU2* marker gene. Yeast transformation was performed using a lithium acetate (LiAc)-based transformation. Empty yeast strains were grown overnight at 30 °C and 180 rpm in 1× Yeast Peptone Dextrose Adenine (YPDA) media, (2% (*w*/*v*) Bacto peptone; 1% (*w*/*v*) Bacto yeast extract; 0.003% (*w*/*v*) adenine hemisulfate; 2% (*w*/*v*) glucose pH 2.5). The overnight cultures were diluted to a concentration of OD546 0.2–0.4 with 1× YPDA and regrown at 30 °C to a final concentration of OD_546_ 0.6–0.8. Cultures were centrifuged for 5 min at 2.500 g, and pellets were resuspended in 2.5 mL sterile water, respectively. 100 μL resuspended yeast cells were added to a polyethylene glycol (PEG)/LiAc mastermix (240 μL 50% PEG, 36 μL 1M LiAc 2xH2O, 2 μL carrier DNA (Clontech)) and 250–600 ng of the appropriate plasmids. Samples were mixed and incubated at 42 °C for 45 min. After the incubation, yeast cells were collected by centrifugation (5 min at 700 g), and pellets were resuspended in 100 μL 0.9% (*w*/*v*) NaCl. Cells were then plated on the appropriate synthetic defined (SD) dropout media to select for transformants, including SD-Trp media (0.67% (*w*/*v*) yeast nitrogen base without amino acids; 0.074% (*w*/*v*)—Trp DO supplement; 2% (*w*/*v*) Bacto agar; 2% (*w*/*v*) glucose, pH 5.8) and SD-Leu media (0.67% (*w*/*v*) yeast nitrogen base without amino acids; 0.069% (*w*/*v*)—Leu DO supplement; 2% (*w*/*v*) Bacto agar; 2% (*w*/*v*) glucose, pH 5.8). After three days of growth at 30 °C, one colony from each transformation was picked, streaked on a fresh plate (SD-Trp/SD-Leu), and incubated for two more days at 30 °C. The two-hybrid assay was performed by mating the transformed yeast strains, as described in the Matchmaker^®^ Gold Yeast Two-Hybrid System User Manual (Clontech). A serial 1:10 dilution of the yeast cells was spotted onto the control DDO media (0.67% (*w*/*v*) yeast nitrogen base without amino acids; 0.064% (*w*/*v*)—Leu-Trp DO supplement; 2% (*w*/*v*) Bacto agar; 2% (*w*/*v*) glucose, pH 5.8) and interaction-selective media, including TDO (0.67% (*w*/*v*) yeast nitrogen base without amino acids; 0.069% (*w*/*v*)—Leu-Trp-His DO supplement; 2% (*w*/*v*) Bacto agar; 2% (*w*/*v*) glucose, pH 5.8), DDO/X/AbA (0.67% (*w*/*v*) yeast nitrogen base without amino acids; 0.064% (*w*/*v*)—Leu-Trp DO supplement; 2% (*w*/*v*) Bacto agar; 2% (*w*/*v*) glucose, pH 5.8; 0.004 (*w*/*v*) X-α-Gal; 0.0002% (*w*/*v*) Aureobasidin A), QDO (0.67% (*w*/*v*) yeast nitrogen base without amino acids; 0.060% (*w*/*v*)—Leu-Trp-His-Ade DO supplement; 2% (*w*/*v*) Bacto agar; 2% (*w*/*v*) glucose, pH 5.8), QDO/X (0.67% (*w*/*v*) yeast nitrogen base without amino acids; 0.060% (*w*/*v*)—Leu-Trp-His-Ade DO supplement; 2% (*w*/*v*) Bacto agar; 2% (*w*/*v*) glucose, pH 5.8; 0.004% (*w*/*v*) X-α-Gal) and QDO/X/AbA (0.67% (*w*/*v*) yeast nitrogen base without amino acids; 0.060% (*w*/*v*)—Leu-Trp-His-Ade DO supplement; 2% (*w*/*v*) Bacto agar; 2% (*w*/*v*) glucose, pH 5.8; 0.004% (*w*/*v*) X-α-Gal; 0.0002% (*w*/*v*) Aureobasidin A). An overview of the used media is listed in Table 1. Yeast growth was monitored after four to five days at 30 °C, and plates were scanned using an Epson Perfection V700 Photo Scanner (Epson Europe B.V., Amsterdam, The Netherlands).

### 4.2. Protein Extraction from Yeast Cells and Western Blot Analysis

To confirm protein expression, yeast cells grown on DDO plates were inoculated into 4.5 mL DDO medium and grown overnight at 30 °C while being shaken (180 rpm). For protein extraction, overnight cultures were centrifuged (5 min at 13,000 rpm), and pellets were resuspended in 100 μL deionized water, respectively. An amount of 100 μL 0.2 M NaOH was added, and samples were incubated for 5 min at room temperature (RT). Samples were centrifuged (5 min at 13,000 rpm), and pellets were resuspended in 30 μL sodium dodecyl sulfate (SDS) sample buffer (0.06 M Tris-HCl pH 6.8; 5% (*v*/*v*) glycerol; 2% (*w*/*v*) SDS). Protein concentration was determined using Bradford Roti-Quant (Roth) according to the manufacturer’s protocol. In total, 20 μg total protein of each sample was diluted with 3× Laemmli buffer (3.4% (*w*/*v*) SDS; 62.5 mM Tris pH 6.8; 10% (*v*/*v*) glycerol; 0.075% (*w*/*v*) bromophenol blue; 5% (*v*/*v*) β-mercaptoethanol). Proteins were denatured by incubation at 95 °C for 5 min. Protein samples were separated on a 10–12% SDS-polyacrylamide gel electrophoresis (PAGE, 20V, 90 min); 1× SDS-running buffer was used (25 mM Tris, 200 mM glycine, 0.1% (*w*/*v*) SDS, pH 8.3). Proteins were transferred to a polyvinylidene difluoride (PVDF) membrane (Roth) using semi-dry transfer (Peqlab; 300 mA, 1 h). Membranes were blocked using 3% (*w*/*v*) milk powder (Sucofin) in 1× TBS-T (25 mM Tris; 137 mM NaCl; 0.1% (*v*/*v*) Tween-20, pH 7.6) for 1 h at RT or overnight at 4 °C. After membranes were washed for 5 min with 1× TBS-T, membranes were incubated for 1 h with the primary antibodies in 1.5% (*w*/*v*) milk powder in 1× TBS-T. Antibodies against GAL4-BD and GAL4-AD, respectively, were used. After being washed three times with 1× TBS-T, membranes were incubated with the corresponding secondary antibody (goat anti-mouse horseradish peroxidase conjugates) followed by another round of washing. After applying luminol (Bio-Rad Laboratories Inc., Hercules, CA, USA) to the membranes, signals were detected using an Amersham Imager600 (GE Healthcare, Chicago, IL, USA). Images were processed with Adobe Photoshop CS5 (Adobe Inc., San José, CA, USA) for adjustment of brightness and contrast.

### 4.3. Protoplast Transformation

Protoplasts were prepared from a root cell culture of *Arabidopsis thaliana* ecotype Col-0 and transformed as described in [40]. Protoplasts were transiently transformed with different concentrations of the respective plasmid DNA; for details, also see https://uni-tuebingen.de/fakultaeten/mathematisch-naturwissenschaftliche-fakultaet/fachbereiche/zentren/zentrum-fuer-molekularbiologie-der-pflanzen/research/central-facilities/plant-transformation/.

### 4.4. MUSCLE Alignment of TIFY/ZIM Domains

Domain sequences of TIFY8 and PPD1 and 2, as well as those of the JAZ proteins, were taken from TAIR using the database HMMSMART. All sequences were aligned according to the multiple alignment tool MUSCLE using CLC Main Workbench 8.1.3 (QIAGEN, Aarhus, Denmark).

### 4.5. Bimolecular Fluorescence Complementation (BiFC), Cytometry, and Confocal Microscopy

Ratiometric BiFC assays were performed to study the homo- and heteromeric interaction of TIFY8 and REV as well as REV interaction with truncated versions of TIFY8 (see Figure 1). Therefore, a single vector which carries a red fluorescent protein (RFP) gene as the expression control as well as both candidate genes which were cloned simultaneously to the N- or the C-terminal part of the yellow fluorescent protein (YFP), respectively, was used. The expression of the fusion proteins is controlled by the 35S promoter in the pBiFCt-2in1-NN vector [41]. For protoplast transfection, 4 µg of the plasmid DNA was used to express the fusion proteins. If the proteins interact with each other, YFP fluorescence is restored by bringing the YFP-N and YFP-C parts together. Interactions were visualized 1 day after transfection by flow cytometry using CytoFLEX (Beckman Coulter, Brea, CA, USA). Both the internal mRFP and any reconstituted YFP were excited by the onboard 488nm laser. Peak emission was captured for YFP in FL1 (525/40 nm) and for RFP in FL3 (610/20 nm). All experiments were performed independently at least 4 times. To detect and localize the interaction in the cells, transfected tobacco leaves were analyzed using confocal microscopy (LSM880, Zeiss, Oberkochen, Germany). Therefore, *Nicotiana benthamiana* plants were cultivated and infiltrated with an *Agrobacteria tumefaciens* suspension, which contained the above-mentioned pBiFCt-2in1 constructs. In total, 500 mL of the bacteria overnight culture was inoculated into fresh LB media with the respective antibiotics and incubated for 4–6 h. This culture was centrifuged at 4000 rpm for 10 min. The pellet obtained was diluted in infiltration media (10 mM MgCl2; 0.5M MES; 100 mM Acetosyringone) to an OD600 of 0.5. Leaves of 4-week-old plants were infiltrated by manual injection with a 1-mL needleless syringe. Imaging was performed 2 days later. At least 3 leaves of different plants were analyzed under a Zeiss LSM 880 Airyscan confocal microscope by using the preset sequential scan settings for YFP (Ex: 514 nm, Em: 517–553 nm) and for RFP (Ex: 561 nm, Em: 597–625 nm). The experiment was performed independently 3 times.

### 4.6. ß-Glucuronidase Reporter Assays

Arabidopsis protoplasts were transformed using 5 µg of effector (pJAN33) and 5 µg of the reporter (pBGWFS7) plasmid DNA. A luciferase construct (pBT8-35SLUCm3) was co-transfected as an internal transformation control. After incubation overnight at 20 °C in darkness, GUS activity assays were performed with the protoplasts, as described by [42]. The basal GUS level at 0min was subtracted from the values of the GUS activity after 2 h of incubation at 37 °C. To correct transformation efficiency, GUS activity was normalized to luciferase fluorescence. As effectors, we analyzed either REV and TIFY8 or a combination of both. Therefore, the coding sequences were cloned into pJAN33. As a reporter, a 2759-bp sequence upstream of the start codon of *WRKY53* was cloned into the binary vector pBGWFS7.0. The JA GUS assays were performed as described above, except that 40 µM JA or the same volume of water was added before overnight incubation.

### 4.7. Plant Cultivation and Plant Lines

*Arabidopsis thaliana* plants were grown on standard soil under short- or long-day conditions. Long-day conditions included 16 h of light; short-day conditions included 8 h of light with only moderate light intensity (60–100 μmol s^−1^ m^−2^) in a climatic chamber at an ambient temperature of 20 °C. Individual leaf positions within the rosette were color-coded according to their age [27]. Plant material was harvested always at the same time of the day to avoid circadian effects. In all experiments, *A. thaliana* Ecotype Columbia-0 was used as wildtype control. The mutant lines used were as follows: *rev5* (EMS mutant; A260V), *tify8-1T* (GK_738B03), *tify8-2T* (SAIL_409_A07), *tify8-3* (CRISPR-CAS 2959-3-4-20/1), *tify8-4* (CRISPR-CAS 2960-21-7-33/3). For CRISPR/Cas9 constructs, design, cloning, genotyping, and selection of homozygous lines were as described [26]. In brief, we designed sgRNA69, sgRNA45, and sgRNA36 to target exon 3, 4, and 5, respectively. Spacers were cloned in pMR218 (sgRNA36) and pMR217 (sgRNA69 and sgRNA45) via a cut–ligation reaction of annealed oligonucleotides with BbsI (Table 2). Vectors were recombined using Gateway in pDE-Cas9 and transformed in Arabidopsis using floral dip. Plants were genotyped each generation using Sanger sequencing and TIDE (Table 2). Finally, two lines were obtained: *tify8-3* (2959-3-4-20/1, with −1;−1 at sgRNA36 and +1;+1 at sgRNA45) and *tify8-4* (2960-21-7-33/3, with −1;−1 at sgRNA36 and −1;−1 at sgRNA69) (Figure 5).

### 4.8. Phenotyping

To analyze onset and progression of senescence in different plant lines, we analyzed a variety of parameters once a week from week 4 to week 8. For this purpose, the corresponding color-coded leaves were used to analyze specific parameters as described in [27]. Before leaves were harvested, the number of leaves, the size of the stems, and the time point of bolting and flowering were determined. Leaf colors were quantified via the automated colorimetric assay (ACA). Electrolyte leakage was measured in leaf No. 4 using a conductivity meter (CM100-2, Reid and Associates, Durban, South Africa). In leaf No. 5 and leaf No. 10, first the activity of the photosystem II (PSII) was assessed by Fv/Fm values using the Imaging-PAM chlorophyll fluorometer (Maxi version, v2-46i, Walz GmbH, Effeltrich, Germany), and subsequently, the chlorophyll was extracted. For qRT-PCR of senescence-associated marker genes, total RNA was extracted from leaves No. 6 and 7. Lipid peroxidation measurements were performed using leaf No. 9. All methods are described in detail in [27]. All phenotyping experiments were performed with a minimum of 6 biological replicates and were independently performed at least three times. All raw data of the phenotyping experiments except for ACA are provided in Appendix A.

### 4.9. Gene Expression Analyses Using qRT-PCR

Total RNA was extracted with the GeneMATRIX Universal RNA Purification Kit (EURx). Subsequent cDNA synthesis was performed with RevertAid Reverse Transcriptase (Thermo Fisher Scientific Inc., Waltham, MA, USA) using oligo-dT primers. For the qRT-PCR, KAPA SYBR^®^ Fast Bio Rad iCycler (Bio-Rad Laboratories Inc., Hercules, CA, USA) and Master Mix was used following the manufacturer´s protocol. For calculation, we used the ∆∆CT method according to [43], in which the expression of the analyzed genes was normalized to *ACTIN2* and set in % of *ACTIN2*. *ACTIN2* has been characterized as suitable reference gene for senescence [44] and is used in many studies that analyze gene expression during senescence, not only in Arabidopsis, but also in other plant species. A list of all primers used can be found in Table 3.

### 4.10. Jasmonic Acid Treatment

Three-week-old Col-0 plants and five-week-old Col-0 plants as well as rev5, tify8-1T, and tify8-4T mutants were sprayed with 100 µM MeJA in DMSO and 0.01% Silwet L-77 every 24 h for 96 h. The second set of plants was treated with the corresponding MOCK solution as a control. Three replicates of each line and time point (6 h, 24 h, and 96 h) were generated. For each replicate, leaves No. 4, 5, and 6 were pooled, and RNA was extracted for quantitative RT-PCR.

### 4.11. Jasmonic Acid Measurements

Four- to seven-week-old Col-0 plants, as well as *rev5*, *tify8-1T*, *tify8-4*, and *TIFY8-OE* plants, were used to determine the JA contents over the development of the different plant lines. For each line and time point, 4 replicates were analyzed. For each replicate, always the same number of punches of the Leaves No. 5 to No. 9 were pooled, and 50 mg (±10%) of leaf material was then analyzed per sample. The frozen sample was retched (5 mm ceramic ball; 30 s) with intermittent cooling. The retched plant material was extracted with 200 µL 80% Methanol (MeOH), which contained 200 nm of D5 JA as a control. The obtained supernatant was transferred to a precooled fresh tube, and the obtained pellet was re-extracted with 200 µL of H_2_O with 0.1% Formic Acid (FA, H_2_CO_2_). Then, this supernatant was combined with the previously transferred 80% MeOH fraction and thoroughly mixed. Both extraction processes included a 5 min ultra-sonic bath at RT, followed by a centrifugation step (5 min, 4 °C, 14,000 rpm). Subsequently, another centrifugation step (10 min. 4 °C, 14,000 rpm) with the combined supernatants was performed. For the analysis of the phytohormone JA, the final supernatant was used directly. An amount of 100 µL of the sample was pipetted into a vial and diluted with 100 µL of H_2_O and 0.1% formic acid. The LCMS profiling analysis was performed using a Micro-LC M5 (Trap and Elute) and a QTRAP6500+ (Sciex) operated in MRM mode. For all MRMs (Ja (1) Q1/Q3 209.1/59, Ja (2) Q1/Q3 209.1/165.1, D5 Ja (1) Q1/Q3 214.1/62, D5 Ja (2) 214.1/170.1) a declustering potential of DP-40, a collision energy of CE-20, and a Dwell time of 5 ms were applied. Chromatographic separation was achieved on a Luna Omega Polar C18 column (3 μm; 100 Å; 150 × 0.5 mm; Phenomenex, Aschaffenburg, Germany) and a Luna C18(2) trap column (5 μm; 100 Å; 20 × 0.5 mm; Phenomenex) with a column temperature of 55 °C. The following binary gradient was applied for the main column at a flow rate of 28 µL min^−1^: 0–0.2 min, isocratic 90% A; 0.2–2 min, linear from 90% A to 30% A; 2–4.5 min, linear from 30% A to 10% A; 4.5–5 min, linear from 10% A to 5% A; 5–5.3 min, isocratic 5% A; 5.3–5.5 min, linear from 5% A to 90% A; 5.5–6 min, isocratic 90% A (A: water, 0.1% aq. formic acid; B: acetonitrile, 0.1% aq. formic acid). The samples were concentrated on the trap column using the following conditions: flow rate 50 µL min^−1^: 0–1.5 min isocratic 95% A; 1.5 min start main gradient; 1.5–1.7 min isocratic 95% A. The injection volume was 50 μL. Analytes were ionized using an Optiflow Turbo V ion source equipped with a SteadySpray T micro electrode (10–50 μL min^−1^) in negative (ion spray voltage: −4500 V) ion mode. The following additional instrument settings were applied: nebulizer and heater gas, nitrogen, 25 and 45 psi; curtain gas, nitrogen, 30 psi; collision gas, nitrogen, medium; source temperature, 200 °C; entrance potential, ±10 V; collision cell exit potential, ±25V. The JA content in a sample was normalized against the D5 Ja values.

### 4.12. Statistical Analyses

All analyses were performed using IBM SPSS Statistics Software (IBM Corp. Released 2021. IBM SPSS Statistics for Windows, Version 28.0. Armonk, NY, USA: IBM Corp.). Comparisons of mean trait values between the different lines were performed using a one-way between-subjects ANOVA. The one-way ANOVA is the simplest case of ANOVA test and is used to compare the mean of multiple groups. If the average variation between groups is large enough compared to the average variation within groups, then it can be concluded that at least one group mean is not equal to the others. A *p*-value of *p* ≤ 0.05 was used in all analyses.

## Figures and Tables

**Figure 1 ijms-24-03079-f001:**
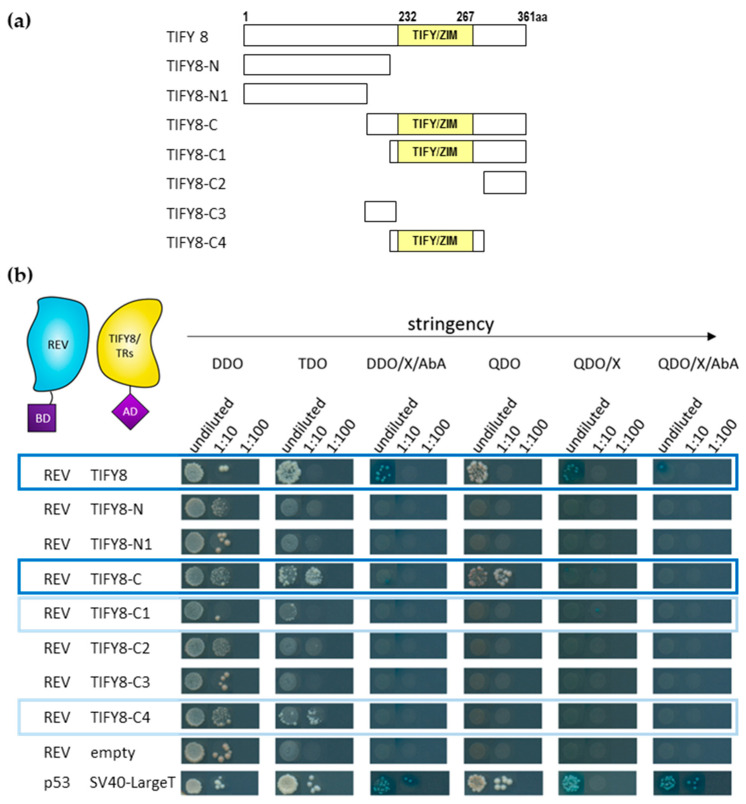
Yeast two-hybrid interactions between REV and TIFY8 and truncated proteins. (**a**) Scheme of TIFY8 and the truncated versions used in the yeast two-hybrid assay. The yellow box represents the TIFY/ZIM domain. TIFY8 full length (1 to 361aa), TIFY8-N (1 to 229aa), TIFY8-N1 (1 to 176aa), TIFY8-C (176 to 361aa), TIFY8-C1 (230 to 361aa), TIFY8-C2 (284 to 361aa), TIFY8-C3 (176 to 229aa), and TIFY8-C4 (239 to 283aa). (**b**) Representative yeast two-hybrid assay between GAL4-BD-REV and TIFY8, as well as a series of truncated versions of the TIFY8 protein shown in (**a**), fused with GAL4-AD. A serial 1:10 dilution of each transformed yeast was spotted onto control (DDO) and different protein–protein interaction selective media with increasing stringency. Blue boxes indicate interactions, and light blue boxes indicate weak interactions.

**Figure 2 ijms-24-03079-f002:**
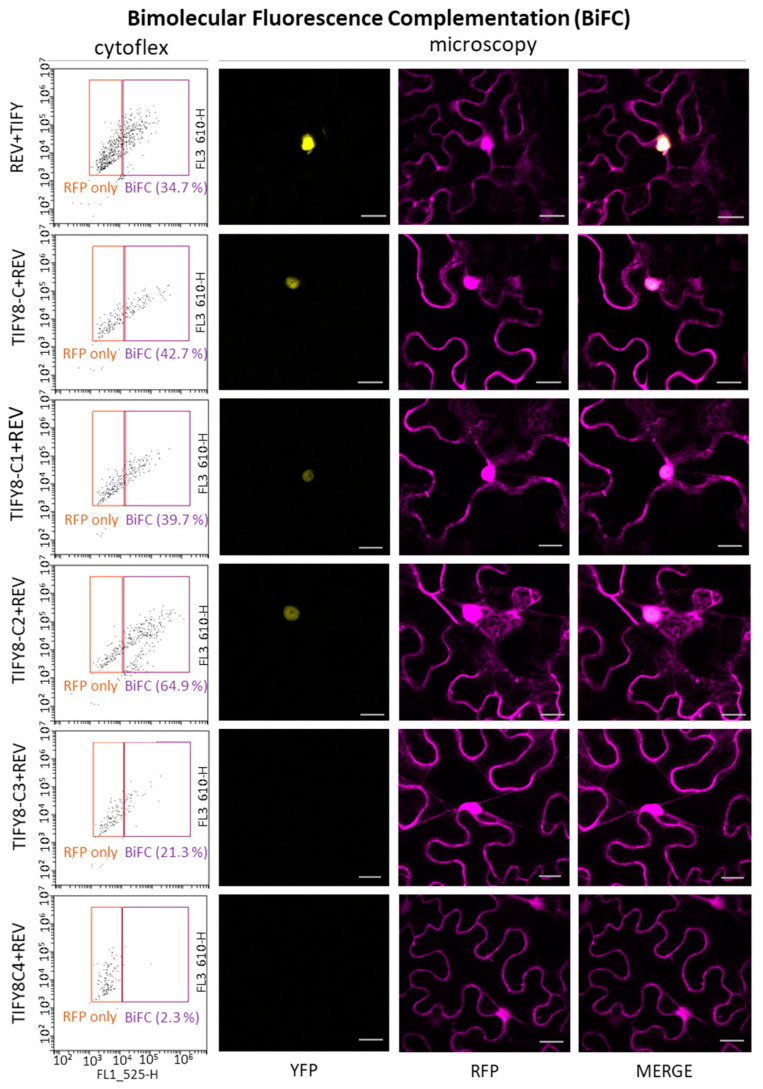
In planta protein–protein interaction between TIFY8 or TIFY8-truncated versions (see Figure 1a) with REV using BiFC in transiently transformed Arabidopsis protoplasts or tobacco leaves. Protoplasts were analyzed with the cytoflex cell sorter (**left**), orange squares indicate transformed protoplasts (RFP), and purple squares indicate interaction via BiFC (YFP). Transiently transformed *Nicotiana benthamiana* leaves were analyzed under the confocal laser scanning microscope (**right**). YFP indicates BiFC; RFP is used as a transformation control. Scale bar indicates 20 µm.

**Figure 3 ijms-24-03079-f003:**
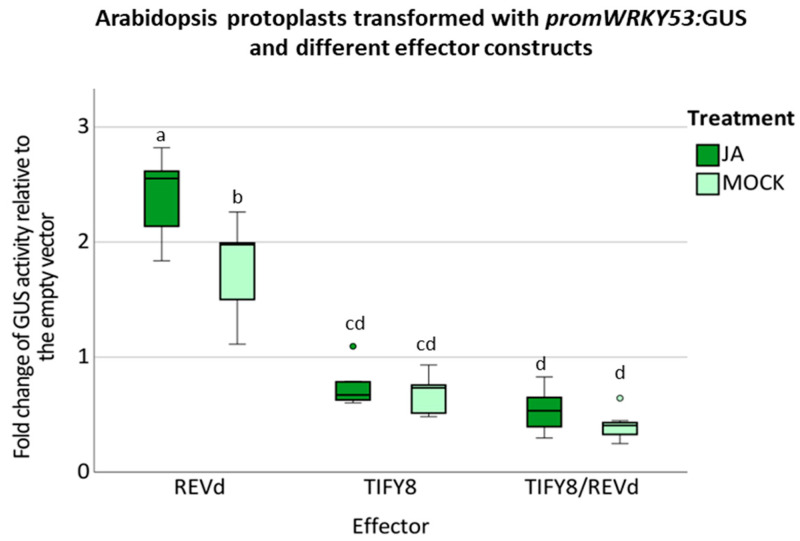
Arabidopsis protoplasts were transiently transformed with a 2.8-kbp-fragment of the *WRKY53* promoter fused to the *GUS* gene as a reporter construct; 35S: *REVd* and 35S: *TIFY8* constructs were used as effector plasmids. These transfected protoplasts were simultaneously incubated with 40 µM JA or the same volume of water for the MOCK condition. A boxplot of the values relative to the empty vector control is presented (*n* = 7). One-way ANOVA test was performed, lowercase letters indicate significant differences among groups (*p* ≤ 0.05).

**Figure 4 ijms-24-03079-f004:**
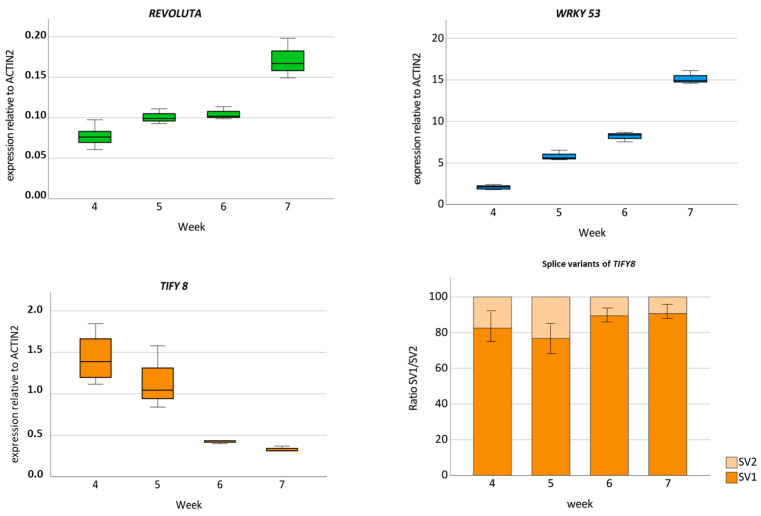
Boxplots presenting the expression of *REVOLUTA*, *WRKY53,* and *TIFY8* over time in leaf tissue of Arabidopsis wildtype plants Col-0. Leaves No. 6 and 7 of each rosette were harvested from 4- to 7-week-old plants. Expression levels were determined via qRT-PCR, and values were normalized to *ACTIN2* (*n* = 3). In the case of *TIFY8*, the ratio for the two existing splicing variants (SV1, SV2, see Appendix A) was determined.

**Figure 5 ijms-24-03079-f005:**
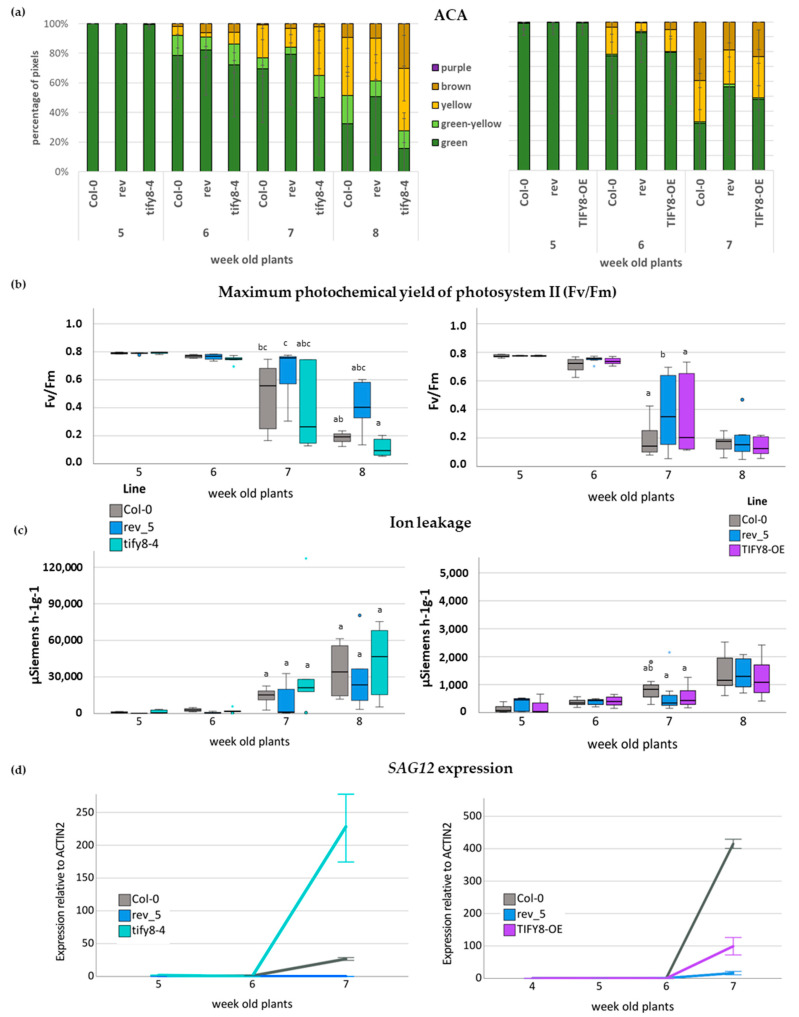
Photosynthetic parameters analyzed for the senescence phenotyping of *tify8-4* and *TIFY8*-OE compared to *rev5* mutants and wildtype Col-0 plants. (**a**) The Automated Colorimetric Assay (ACA) to categorize the color of individual leaves of at least six plants pixelwise into five categories: green, green-yellow, yellow, brown, and purple. The percentage of each group with respect to total pixel number of all leaves (1–10) is presented (*n* = 6). (**b**) Boxplot of Fv/Fm values measured with PAM for leaves No. 5 of 4- to 7-week-old plants (*n* = 6–8). (**c**) Boxplot of the decrease in solute retention determined through ion leakage in leaves No. 4 of 4- to 7-week-old plants (*n* = 6–8). One-way ANOVA test was performed, lowercase letters indicate significant differences among groups (*p* ≤ 0.05). (**d**) Gene expression of the senescence-associated marker genes SAG12 was analyzed by qRT-PCR and normalized to the expression of the *ACTIN2* gene (mean values ± SD, *n* = 3).

**Figure 6 ijms-24-03079-f006:**
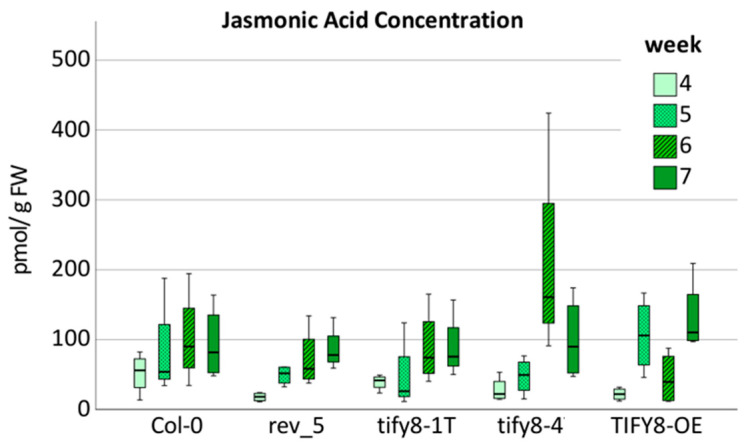
Boxplot of JA concentrations measured by LCMS in Col-0, *rev5, tify8-1T*, and *tify8-4* mutants and *TIFY8*-OE plants. The concentration was determined in pools of leaves No. 5 to 9 of 4- to 7-week-old-plants (*n* = 4).

**Figure 7 ijms-24-03079-f007:**
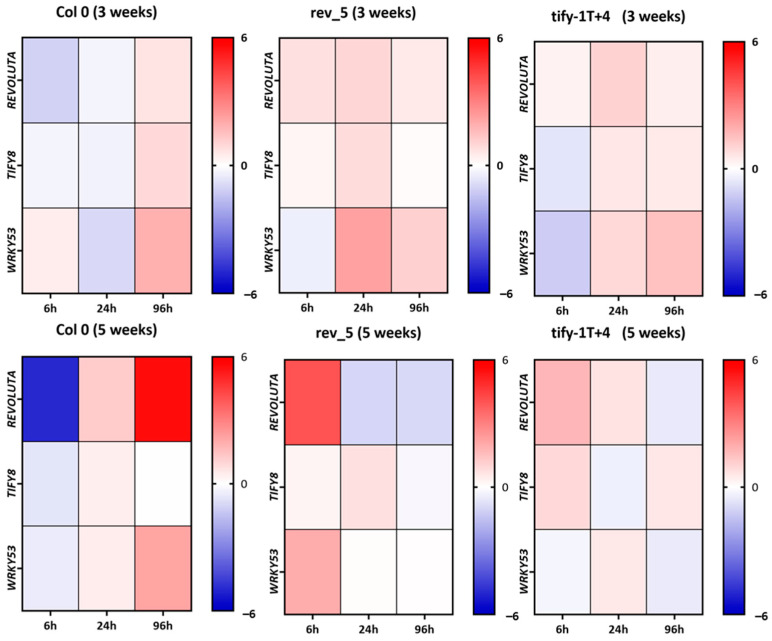
Heat map of the gene expression measured by qRT-PCR of *REV*, *TIFY8,* and *WRKY53* in Col-0, *rev5,* and *tify8* mutant plants for 3-week-old and 5-week-old plants 6 h, 24 h, and 96 h after JA treatment. Expression is shown as log2-fold changes relative to the respective MOCK treatments; blue color indicates repression, whereas red color indicates induction.

**Figure 8 ijms-24-03079-f008:**
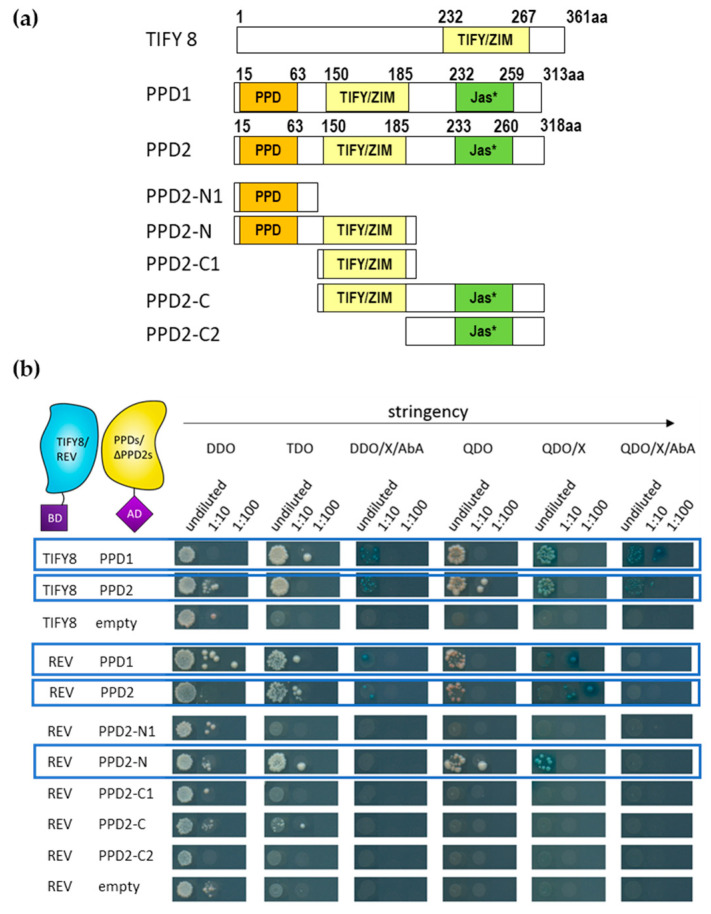
Yeast two-hybrid interactions between PEAPODs, PPD2-truncated proteins, TIFY8, and REV. (**a**) Scheme of the PPD1, PPD2, TIFY8, and truncated PPD2 versions used in the yeast two-hybrid assay. The orange box represents the PEAPOD domain (PPD), the yellow box represents the TIFY/ZIM domain, and the green box represents the Jas-like domain (Jas*). PPD2 full length (1 to 318aa), PPD2-N1 (1 to 117aa), PPD2-N (1 to 204aa), PPD2-C1 (117 to 204aa), PPD2-C (117-316aa), and PPD2-C2 (205 to 316aa). (**b**) Representative yeast two-hybrid assay between GAL4-BD-TIFY8 or GAL4-REV and PPDs, as well as a series of truncated versions of the PPD2 protein shown in (**a**) fused with GAL4-AD. A serial 1:10 dilution of each transformed yeast was spotted onto control (DDO) and different protein-protein interaction selective media with increasing stringency. Blue boxes indicate interactions.

**Figure 9 ijms-24-03079-f009:**
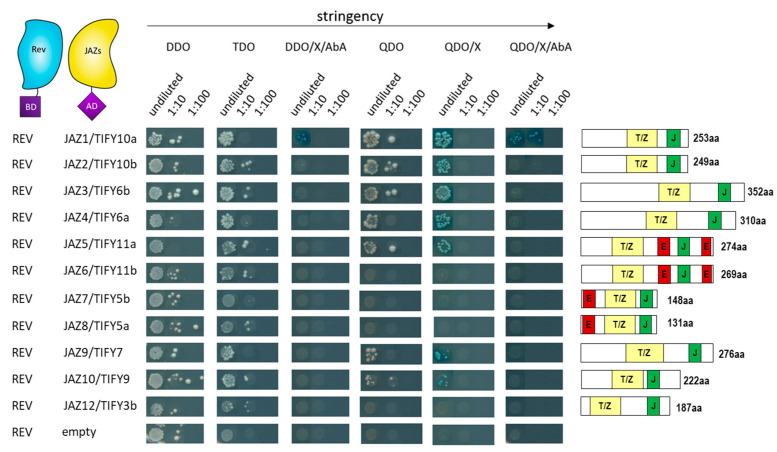
*Yeast two-hybrid interactions between REV and JAZ proteins*. Representative yeast two-hybrid assay between Gal4-BD-REV and GAL4-AD-JAZ fusion proteins. A serial 1:10 dilution of each transformed yeast was spotted onto control (DDO) and different protein–protein interaction-selective media with increasing stringency. Blue boxes indicate interactions. Protein domains of the different JAZ proteins are indicated on the right. The yellow box represents the TIFY/ZIM domain. The green box represents the Jas domain. The red box indicates the repressing EAR domain.

**Figure 10 ijms-24-03079-f010:**
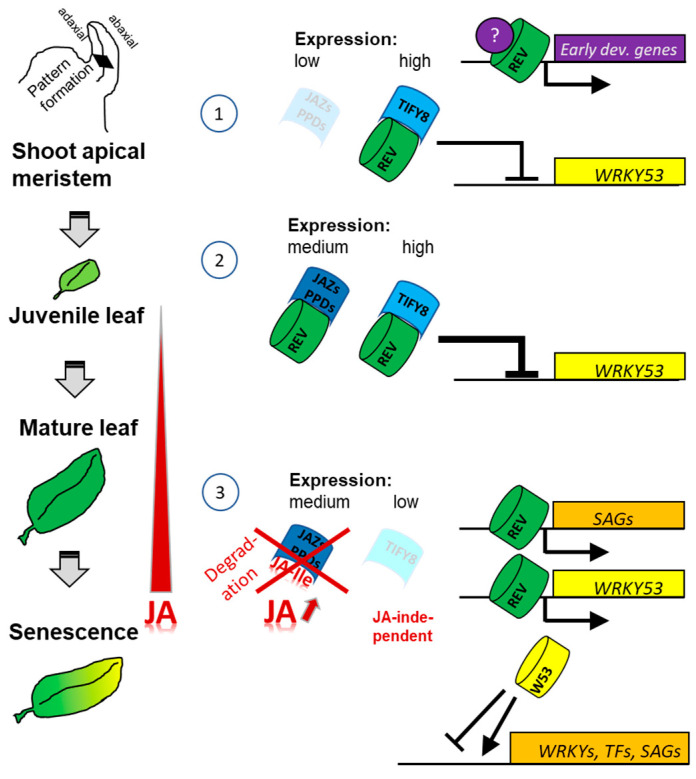
Model of the development-specific regulatory effects of TIFY8, PPDs, and JAZs on REV. (**1**) TIFY8, which is highly expressed during early development, blocks *WRKY53* expression by its interaction with REV. How REV activates other early developmental genes is still unclear and needs most likely additional factors or modifications. In this stage, *JAZ* as well as *PPD* expression is low. (**2**) In juvenile leaves, *JAZs* and *PPDs* are expressed, but JA levels are still low; therefore, even enhanced repression of *WRKY53* is achieved by the additional complex formation of JAZs/PPDs with REV. (**3**) At the transition from maturation to senescence, JA levels increase, and JA-Ile mark the JAZs and PPDs for degradation via the 26S proteasome through the interaction with the F-box protein COI1. However, TIFY8 has no Jas domain and cannot be marked for degradation by JA. Instead, expression of the *TIFY8* gene is strongly reduced. Both the JA-dependent and the JA-independent pathways lead to an activation of gene expression of REV target genes, including *WRKY53,* which then activates other *WRKY*s, other *TF*s, and additional *SAG*s. TF: transcription factors; SAGs: senescence-associated genes.

**Table 1 ijms-24-03079-t001:** Media used for the yeast two-hybrid assay.

Function	Name	Description
vector selection	Double dropout (DDO)	SD/-Trp/-Leu
interaction	Triple dropout (TDO)	SD/-Trp/-Leu/-His
	Quadruple dropout (QDO)	SD/-Trp/-Leu/-His/-Ade
	QDO/X	SD/-Trp/-Leu/-His/-Ade supplemented with X-α-Gal (X)
	QDO/X/AbA	SD/-Trp/-Leu/-His/-Ade supplemented with X-α-Gal (X) and Aureobasidin A (AbA)

**Table 2 ijms-24-03079-t002:** Primers for CRISPR/Cas9.

Oligonucleotides uses for cloning spacers			
LAPAU*3124	ATTGCAAACCAGCCTCCACGCGG	Fw	sgRNA69
LAPAU3125	AAACCCGCGTGGAGGCTGGTTTG	Rv	sgRNA69
LAPAU3126	ATTGCTTGACCGCCATAGAAGA	Fw	sgRNA45
LAPAU3127	AAACTCTTCTATGGCGGTCAAG	Rv	sgRNA45
LAPAU3128	ATTGCCTTGGCAGGATCAAGCGG	Fw	sgRNA36
LAPAU3129	AAACCCGCTTGATCCTGCCAAGG	Rv	sgRNA36
Genotyping primers			
CROPGEN *68	TCACTTCACGACTCAGGAGC	Fw	Genotype sgRNA 69
CROPGEN69	CCATTATCACATCCGCCTGC	Rv	Genotype sgRNA 69
CROPGEN70	AACAGGGATGAAAGGTCCCG	Fw	Genotype sgRNA 45 or 36
CROPGEN71	AGACCTGATTACTCTACTCCACTCA	Rv	Genotype sgRNA 45 or 36

* LAPAU and CROPGEN are internal names and are not meaningful.

**Table 3 ijms-24-03079-t003:** Primers for qRT-PCR.

Gene Name	Accession Number	Primer Sequence (for/rev)
Phenotyping *ACTIN2*	At3g18780	ACCCGATGGGCAAGTCATCACG TCCCACAAACGAGGGCTGGA
*SAG12*	At5g45890	GCTTTGCCGGTTTCTGTTG GTTTCCCTTTCTTTATTTGTGTTG
*SAG13*	At2g29350	AGGGAGCATCGTGCTCATATCC CCAGCTGATTCATGGCTCCTTTG
Development and *ACTIN2*	MeJA treatment At3g18780	AAGCTCTCCTTTGTTGCTGTT GTTGTCTCGTGGATTCCAGCAGCTT
*TIFY8*	At4g32570	CCGACAGACAGAACAAGATAAGC AAGCAGAAGCCGTGGAAGG
*REVOLUTA*	At5g60690	TCAGCTTGTCTGCGAAAATG ACCCAATCAACAGCAGTTCC
*WRKY53*	At4g23810	CAGACGGGGATGCTACGG GGCGAGGCTAATGGTGGT
Splicing variants *TIFY8-SV1*	At4g32570	TGTATGAAGGAGGCAGCTCTAAG TCATGTGGCTTCTTTTTCAGGATC
*TIFY8-SV2*	At4g32570	TGTATGAAGGAGGCAGCTCTAAG TCAGTATTTGTAAGAAGCTAACCA

## Data Availability

Not applicable.

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
