# Peer review of "The Non-JAZ TIFY Protein TIFY8 of Arabidopsis thaliana Interacts with the HD-ZIP III Transcription Factor REVOLUTA and Regulates Leaf Senescence"

_ijms, 2023, doi:10.3390/ijms24043079_

Round 1

Reviewer 1 Report

The manuscript by Andrade et al. studies the interaction of the non-JAZ TIFY protein TIFY8 of Arabidopsis with the HD-ZIP III transcription factor REVOLUTA and its implication on leaf senescence

The work makes an interesting contribution, complementing previous studies of the group, to the understanding of the role of the Transcription factor REV on regulation of SAG genes.

In this work the authors characterized the protein-interaction partners of REV that could regulate senescence-specificity. The interaction between REV and the TIFY family member TIFY8 was evaluated by yeast two-hybrid assays as well as by bimolecular fluorescence complementation in planta, both in Arabidopsis protoplasts and Nicotiana benthamiana leaves. Based on these assays authors showed that interaction inhibited REV´s function as an activator of WRKY53 expression.  They also studied the effect of Jasmonic acid (JA) on TIFY8 and REV expression REV. In addition they also studied the interaction with other members of the TIFY family as PEAPODs and  JAZ

The work makes an interesting contribution to the plant scientific community, due to the relevance of senescence in plant development and environmental response. Although, some items should be addressed, either by completing or clarifying different aspects or concepts listed below, before the manuscript can be recommended for publishing.

 Comments to be addressed

 Material and Methods:  although it is mentioned in the phenotyping experiments the number of assays as well as the number of biological replicas used, the data of the different assays and statistical analysis should be included in supplementary material as the Figures showed average data

 Include a paragraph with the statistical analysis and tests used along the work. Even when there are references to the methods used in Figure legends, they should be described properly in this section.

Results

Authors performed expression analysis by qRT-PCT using ACTIN as reference genes (Results 2.3, 2.4, 2.5). Is is well documented in literature the necessity of evaluating different reference genes for expression studies based on real time PCR, in order to have robust results, taking into account that putative reference genes are not stable under different experimental conditions. If previous evaluation were performed under the experimental conditions conducted for expression studies, comments in these aspects should be included.

Discussion

In order to elucidate the complex network involved in senescence regulation through REV regulation and its interaction with different TIFY members, both JA-independent as TIFY8 and JA-dependent through PEAPODs and JAZ, different transcriptional approximations based on RNAseq should be performed and integrative functional studies analyzed. In the discussion section it will important to discuss this points.

Author Response

Responses are given point by point in bold

Reviewer 1:

The manuscript by Andrade et al. studies the interaction of the non-JAZ TIFY protein TIFY8 of Arabidopsis with the HD-ZIP III transcription factor REVOLUTA and its implication on leaf senescence

The work makes an interesting contribution, complementing previous studies of the group, to the understanding of the role of the Transcription factor REV on regulation of SAG genes.

In this work the authors characterized the protein-interaction partners of REV that could regulate senescence-specificity. The interaction between REV and the TIFY family member TIFY8 was evaluated by yeast two-hybrid assays as well as by bimolecular fluorescence complementation in planta, both in Arabidopsis protoplasts and Nicotiana benthamiana leaves. Based on these assays authors showed that interaction inhibited REV´s function as an activator of WRKY53 expression.  They also studied the effect of Jasmonic acid (JA) on TIFY8 and REV expression REV. In addition, they also studied the interaction with other members of the TIFY family as PEAPODs and JAZ

The work makes an interesting contribution to the plant scientific community, due to the relevance of senescence in plant development and environmental response. Although, some items should be addressed, either by completing or clarifying different aspects or concepts listed below, before the manuscript can be recommended for publishing.

 Comments to be addressed

 Material and Methods:  although it is mentioned in the phenotyping experiments the number of assays as well as the number of biological replicas used, the data of the different assays and statistical analysis should be included in supplementary material as the Figures showed average data.

All raw data of the phenotyping experiments have been added to the Supplementary Material along with a graphical scheme of the leaves used for the different assays.

 Include a paragraph with the statistical analysis and tests used along the work. Even when there are references to the methods used in Figure legends, they should be described properly in this section.

A paragraph to describe the statistical analyses was added to the Material and Method section

Results

Authors performed expression analysis by qRT-PCT using ACTIN as reference genes (Results 2.3, 2.4, 2.5). Is is well documented in literature the necessity of evaluating different reference genes for expression studies based on real time PCR, in order to have robust results, taking into account that putative reference genes are not stable under different experimental conditions. If previous evaluation were performed under the experimental conditions conducted for expression studies, comments in these aspects should be included.

The ACTIN2 gene was analyzed already in early days of qRT-PCR to be a suitable reference gene for senescence analyses and was used ever since in many different senescence studies, not only in Arabidopsis but also in other plant species. Moreover, we have also tested several other reference genes including GAPDH, tubulin, elongation factor alpha and others. However, ACTIN2 was more robust in its expression than all other genes followed by GAPDH. A reference was added to the manuscript.

Reviewer 2 Report

Please find enclosed the revue of the manuscript entitled “The non-JAZ TIFY protein TIFY8 of Arabidopsis thaliana interacts with the HD-ZIP III transcription factor REVOLUTA 3 and regulates leaf senescence."

The aim of the paper is to better understand the regulation of the HD-ZIP III transcription (TF) factor REVOLUTA (REV) during senescence via the interaction with TIFY8 protein. REV is involved in early leaf development and later in leaf senescence. In this later context, REV binds to the promoter of a central regulator of senescence, the TF WRKY53. This interaction is specific of senescence, and based on the authors, the interaction REV-TIFY8 could mediate this specificity. At first the authors use Y2H to confirm REV and TIFY8 protein interaction and identify the TIFY/ZIM domain with the flanking regions of TIFY, as the interaction domain.  By  BiFC, in protoplast and Tobacco leaves, they validate the interaction. By performing transient expression in protoplast using PWRKY53:GUS they show that while REV alone regulates the reporter gene expression, the presence of TIFY8  abolished GUS expression. Therefore they suggest that TIFY8 /REV interaction prevent WRKY53 gene expression. This regulation is not affected by JA treatment. This reporter assay was  correlated with gene expression of TIFY8, WRKY53 and REV during leaf development and the authors showed that at the onset of monocarpic senescence development, WRKY53 and REV are well expressed while TIFY8 is down regulated. Using several alleles of tyf8 mutants and plant OE TIFY8, they aimed to understand the biological relevance of this interaction. By using a set of measurement characteristic of leaf senescence (Photosynthetic parameters, expression level of senescence genes) they show that TIFY8 act as a repressor of leaf senescence.  Because JA increased with the age of the plant (which measurement was performed again here) and because REV regulation of PWRKY53:GUS is sentitive to JA in protoplast assays the  authors wanted to analyse the impact of JA on REV/TIFY8/WRKY53 network . For this they measured at 2 stages of leaf development, for WT and mutant, REV/TIFY8/WRKY53 gene expression after JA treatment. The author concluded that JA is involved in a complex and developmental manner in the regulation of the 3 genes. Finally, because TIFY8 has no domain able to sense JA, they evaluate whether REV can bind to other member of the TIFY family, the PEAPOD and JAZ proteins. They identified an interaction by Y2H with TIFY8 and PPD protein and REV with PPD. By Focusing on PPD2 they showed that both domains, the PPD and the TIFY/ZIM domains were required for interaction with REV. They also showed an interaction between REV and specific TIFY/JAZZ member which can sense JA. Therefore the JA dependent regulation of REV could be mediated by PEAPO/JAZZ protein.

The manuscript is very well written. Experiments are well designed and well explained. However I believe the manuscript is very long. Many figures could be fused, legends could be lighten.

Please find below some minor comments.

Introduction:

The introduction is very detailed and maybe paragraphs 2-4 could be reduced.

Results:

There are lot of figures, I wonder whether it is possible to reduce some, mix some, or move some to supplemental.

2.1

In this paragraph, 2 times the authors says they will test truncated versions. Please, remove one. Did the authors test BD-TYF and AD-REV as well?

2.2:

Could the binding of REV to WRKY53 be tested by EMSA? And does TYF8 inhibits REV binding to DNA or do the complex still bind to DNA? Did the author try the protoplast assay with TIFY lacking TIFY/ZIM or the motif alone, in the presence of REV?

2.4:

It would have been nice also to OE in plant a TIFY protein lacking the TIFY/ZYM domain.

Figure 5 could be moved to supplemental.

Figure 6: Would it be possible that this figure presents only one tyf8 mutant? It seems that they are all behaving the same. Then Figure 6 and 7 could be put together.

Figures 8 and 9 should be put together.

2.5

What is the phenotype of a mutant that does not produce JA? in term of senescence. Could these gene expression have been done in a ja mutant? What is impact of JA on the senescence phenotype in the figure 11?

Figure 12 and 13 of Y2H could be put together?

Legends for Y2H are very long and very repetitive. Experimental details should be swapped to material and method.

2.6: did the author check the expression of the construct in Y2H, as they did in figure 1?

Discussion:

I would have appreciated a final figure presenting a model to explain the regulation of REV in early leaf development and during senescence.

Material and Method:

Table II: what does LAPAU3 and CROPGEN Mean?

Line 748: quantified and not quanitified

Table III: what does “for phenotyping “mean in this table??

Lines 780-781:  “retched” does not sound right.

Supplemental Material lines 812-826: please check dot, coma

Author Response

Responses are given point by point in bold

Reviewer 2:

The aim of the paper is to better understand the regulation of the HD-ZIP III transcription (TF) factor REVOLUTA (REV) during senescence via the interaction with TIFY8 protein. REV is involved in early leaf development and later in leaf senescence. In this later context, REV binds to the promoter of a central regulator of senescence, the TF WRKY53. This interaction is specific of senescence, and based on the authors, the interaction REV-TIFY8 could mediate this specificity. At first the authors use Y2H to confirm REV and TIFY8 protein interaction and identify the TIFY/ZIM domain with the flanking regions of TIFY, as the interaction domain.  By  BiFC, in protoplast and Tobacco leaves, they validate the interaction. By performing transient expression in protoplast using PWRKY53:GUS they show that while REV alone regulates the reporter gene expression, the presence of TIFY8  abolished GUS expression. Therefore they suggest that TIFY8 /REV interaction prevent WRKY53 gene expression. This regulation is not affected by JA treatment. This reporter assay was  correlated with gene expression of TIFY8, WRKY53 and REV during leaf development and the authors showed that at the onset of monocarpic senescence development, WRKY53 and REV are well expressed while TIFY8 is down regulated. Using several alleles of tyf8 mutants and plant OE TIFY8, they aimed to understand the biological relevance of this interaction. By using a set of measurement characteristic of leaf senescence (Photosynthetic parameters, expression level of senescence genes) they show that TIFY8 act as a repressor of leaf senescence.  Because JA increased with the age of the plant (which measurement was performed again here) and because REV regulation of PWRKY53:GUS is sentitive to JA in protoplast assays the  authors wanted to analyse the impact of JA on REV/TIFY8/WRKY53 network . For this they measured at 2 stages of leaf development, for WT and mutant, REV/TIFY8/WRKY53 gene expression after JA treatment. The author concluded that JA is involved in a complex and developmental manner in the regulation of the 3 genes. Finally, because TIFY8 has no domain able to sense JA, they evaluate whether REV can bind to other member of the TIFY family, the PEAPOD and JAZ proteins. They identified an interaction by Y2H with TIFY8 and PPD protein and REV with PPD. By Focusing on PPD2 they showed that both domains, the PPD and the TIFY/ZIM domains were required for interaction with REV. They also showed an interaction between REV and specific TIFY/JAZZ member which can sense JA. Therefore the JA dependent regulation of REV could be mediated by PEAPO/JAZZ protein.

The manuscript is very well written. Experiments are well designed and well explained. However I believe the manuscript is very long. Many figures could be fused, legends could be lighten.

Please find below some minor comments.

Introduction:

The introduction is very detailed and maybe paragraphs 2-4 could be reduced.

Paragraphs 2-4 have been shortened. 

Results:

There are lot of figures, I wonder whether it is possible to reduce some, mix some, or move some to supplemental.

Done, see below

2.1

In this paragraph, 2 times the authors says they will test truncated versions. Please, remove one. Did the authors test BD-TYF and AD-REV as well?

Yes, we could also determine interaction but this combination was less efficient most likely due to a very low expression of AD-REV. Therefore, we did all further analyses with BD-REV which appeared to be expressed much better and to a very comparable level.

2.2:

Could the binding of REV to WRKY53 be tested by EMSA? And does TYF8 inhibits REV binding to DNA or do the complex still bind to DNA? Did the author try the protoplast assay with TIFY lacking TIFY/ZIM or the motif alone, in the presence of REV?

These are very good suggestions which we are currently planning and performing in our follow-up experiments.

2.4:

It would have been nice also to OE in plant a TIFY protein lacking the TIFY/ZYM domain.

These are also ongoing experiments, we are also doing complementation lines using the truncated versions in the tify8 mutant background.

Figure 5 could be moved to supplemental.

Done

Figure 6: Would it be possible that this figure presents only one tyf8 mutant? It seems that they are all behaving the same. Then Figure 6 and 7 could be put together.

Figures 8 and 9 should be put together.

Done, we have combined the Figure 6-9 as suggested and moved some parts to the Supplements.

2.5

What is the phenotype of a mutant that does not produce JA? in term of senescence. Could these gene expression have been done in a ja mutant? What is impact of JA on the senescence phenotype in the figure 11?

As we harvested the plants at three different time points after JA treatment, we could not follow up the effect on the senescence phenotype of our plant lines. However, it will be difficult to see, whether JA application would change the senescence phenotype, as the mutants show already a senescence phenotype. As JA has already been discovered to induce senescence by He and coworkers in 2002 (He et al., 2002, Plant Physiology, 128, 876–884, https://doi.org/10.1104/pp.010843), at least in Col-0, we would expect that we can observe premature senescence, but we did not test this.

Figure 12 and 13 of Y2H could be put together?

Done

Legends for Y2H are very long and very repetitive. Experimental details should be swapped to material and method.

Legends have been shortened.

2.6: did the author check the expression of the construct in Y2H, as they did in figure 1?

All expression data have been provided in the Supplemental Figures.

Discussion:

I would have appreciated a final figure presenting a model to explain the regulation of REV in early leaf development and during senescence.

A model has been provided in Figure 10 now.

Material and Method:

Table II: what does LAPAU3 and CROPGEN Mean?

These are only internal nickname: LA-PAU (for Laurens Pauwels) and of the CROP GENOME ENGINEERING FACILITY (Laurens Pauwels team). A footnote has been added.

Line 748: quantified and not quanitified

Corrected

Table III: what does “for phenotyping “mean in this table??

We used different primer sets for different experiments (phenotyping, development and MeJA treatment, splicing variants), this has been added to the table. We hope this has become clearer now.

Lines 780-781:  “retched” does not sound right.

The material was homogenized with a retch mill with the specified ceramic balls, therefore this term is correct.  

Supplemental Material lines 812-826: please check dot, coma

Has been corrected.